# Stable task information from an unstable neural population

**Michael E Rule[1], Adrianna R Loback[1], Dhruva V Raman[1], Laura N Driscoll[2], Christopher D Harvey[3], Timothy O'Leary[1]***

[1]Department of Engineering, University of Cambridge, Cambridge, United Kingdom; [2]Department of Electrical Engineering, Stanford University, Stanford, United States; [3]Department of Neurobiology, Harvard Medical School, Boston, United States

**Abstract** Over days and weeks, neural activity representing an animal's position and movement in sensorimotor cortex has been found to continually reconfigure or 'drift' during repeated trials of learned tasks, with no obvious change in behavior. This challenges classical theories, which assume stable engrams underlie stable behavior. However, it is not known whether this drift occurs systematically, allowing downstream circuits to extract consistent information. Analyzing long-term calcium imaging recordings from posterior parietal cortex in mice (*Mus musculus*), we show that drift is systematically constrained far above chance, facilitating a linear weighted readout of behavioral variables. However, a significant component of drift continually degrades a fixed readout, implying that drift is not confined to a null coding space. We calculate the amount of plasticity required to compensate drift independently of any learning rule, and find that this is within physiologically achievable bounds. We demonstrate that a simple, biologically plausible local learning rule can achieve these bounds, accurately decoding behavior over many days.

## Introduction

A core principle in neuroscience is that behavioral variables are represented in neural activity. Such representations must be maintained to retain learned skills and memories. However, recent work has challenged the idea of long-lasting neural codes (*Rumpel and Triesch, 2016*). In our recent work (*Driscoll et al., 2017*), we found that neural activity–behavior relationships in individual posterior parietal cortex (PPC) neurons continually changed over many days during a repeated virtual navigation task. Similar 'representational drift' has been shown in other neocortical areas and hippocampus (*Attardo et al., 2015*; *Ziv et al., 2013*; *Levy et al., 2019*). Importantly, these studies showed that representational drift is observed in brain areas essential for performing the task long after the task has been learned.

These experimental observations raise the major question of whether drifting representations are fundamentally at odds with the storage of stable memories of behavioral variables (e.g. *Ganguly and Carmena, 2009*; *Tonegawa et al., 2015*). Theoretical work has proposed that a consistent readout of a representation can be achieved if drift in neural activity patterns occurs in dimensions of population activity that are orthogonal to coding dimensions - in a 'null coding space' (*Rokni et al., 2007*; *Druckmann and Chklovskii, 2012*; *Ajemian et al., 2013*; *Singh et al., 2019*). This can be facilitated by neural representations that consist of low-dimensional dynamics distributed over many neurons (*Montijn et al., 2016*; *Gallego et al., 2018*; *Hennig et al., 2018*; *Degenhart et al., 2020*). Redundancy could therefore permit substantial reconfiguration of tuning in single cells without disrupting neural codes (*Druckmann and Chklovskii, 2012*; *Huber et al., 2012*; *Kaufman et al., 2014*; *Ni et al., 2018*; *Kappel et al., 2018*). However, the extent to which drift is confined in such a null coding space remains an open question.

*For correspondence:
tso24@cam.ac.uk

**Competing interests:** The authors declare that no competing interests exist.

Purely random drift, as would occur if synaptic strengths and other circuit parameters follow independent random walks, would eventually disrupt a population code. Several studies have provided evidence that cortical synaptic weights and synaptic connections exhibit statistics that are consistent with a purely random process (*Moczulska et al., 2013*; *Loewenstein et al., 2011*; *Loewenstein et al., 2015*). Indeed, our previous experimental findings reveal that drift includes cells that lose representations of task relevant variables, suggesting that some component of drift affects coding dimensions (*Driscoll et al., 2017*).

Together, these observations raise fundamental questions that have not been directly addressed with experimental data, and which we address here. First, to what extent can ongoing drift in task representations be confined to a null coding space over extended periods while maintaining an accurate readout of behavioral variables in a biologically plausible way? Second, how might we estimate how much additional ongoing plasticity (if any) would be required to maintain a stable readout of behavioral variables, irrespective of specific learning rules? Third, is such an estimate of ongoing plasticity biologically feasible for typical levels of connectivity, and typical rates of change observed in synaptic strengths? Fourth, can a local, biologically plausible plasticity mechanism tune readout weights to identify a maximally stable coding subspace and compensate any residual drift away from this subspace?

We addressed these questions by modelling and analyzing data from *Driscoll et al., 2017*. This dataset consists of optical recordings of calcium activity in populations of hundreds of neurons in Posterior Parietal Cortex (PPC) during repeated trials of a virtual reality T-maze task (*Figure 1a*). Mice were trained to associate a visual cue at the start of the maze with turning left or right at a T-junction. Behavioral performance and kinematic variables were stable over time with some per-session variability (mouse four exhibited a slight decrease in forward speed; *Figure 2—figure supplement 1*). Full experimental details can be found in the original study.

Previous studies identified planning and choice-based roles for PPC in the T-maze task (*Harvey et al., 2012*), and stable decoding of such binary variables was explored in *Driscoll et al., 2017*. However, in primates PPC has traditionally been viewed as containing continuous motor-related representations (*Andersen et al., 1997*; *Andersen and Buneo, 2002*; *Mulliken et al., 2008*), and recent work (*Krumin et al., 2018*; *Minderer et al., 2019*) has confirmed that PPC has an equally motor-like role in spatial navigation in rodents (*Calton and Taube, 2009*). It is therefore important to revisit these data in the context of continuous kinematics encoding.

Previous analyses showed that PPC neurons activated at specific locations in the maze on each day. When peak activation is plotted as a function of (linearized) maze location, the recorded population tiles the maze, as shown in *Figure 1b*. However, maintaining the same ordering in the same population of neurons revealed a loss of sequential activity over days to weeks (top row of *Figure 1b*). Nonetheless, a different subset of neurons could always be found to tile the maze in these later experimental sessions. In all cases, the same gradual loss of ordered activation was observed (second and third rows, *Figure 1b*). *Figure 1c* shows that PPC neurons gain or lose selectivity and occasionally change tuning locations. Together, these data show that PPC neurons form a continually reconfiguring representation of a fixed, learned task.

## PPC representations facilitate a linear readout

We asked whether precise task information can be extracted from this population of neurons, despite the continual activity reconfiguration evident in these data. We began by fitting a linear decoder for each task variable of interest (animal location, heading, and velocity) for each day. This model has the form $x(t) = M^\top z(t)$, where $x(t)$ is the time-binned estimate of position, velocity or heading (view angle) in the virtual maze, $M$ is a vector of weights, and $z(t)$ is the normalized time-binned calcium fluorescence (Materials and methods: Decoding analyses).

Example decoding results for two mice are shown in *Figure 2a*, and summaries of decoding performance for four mice in *Figure 2b*. Position, speed, and view angle can each be recovered with a separate linear model. The average mean absolute decoding error for all animals included in the analysis was 47.2 cm $\pm$8.8 cm (mean $\pm$1 standard deviation) for position, 9.6 cm/s $\pm$2.2 cm/s for speed, and 13.8° $\pm$ 4.0° for view angle (Materials and methods: Decoding analyses).

We chose a linear decoder specifically because it can be interpreted biologically as a single 'readout' neuron that receives input from a few hundred PPC neurons, and whose activity approximates a linear weighted sum. The fact that a linear decoder recovers behavioral variables to reasonable

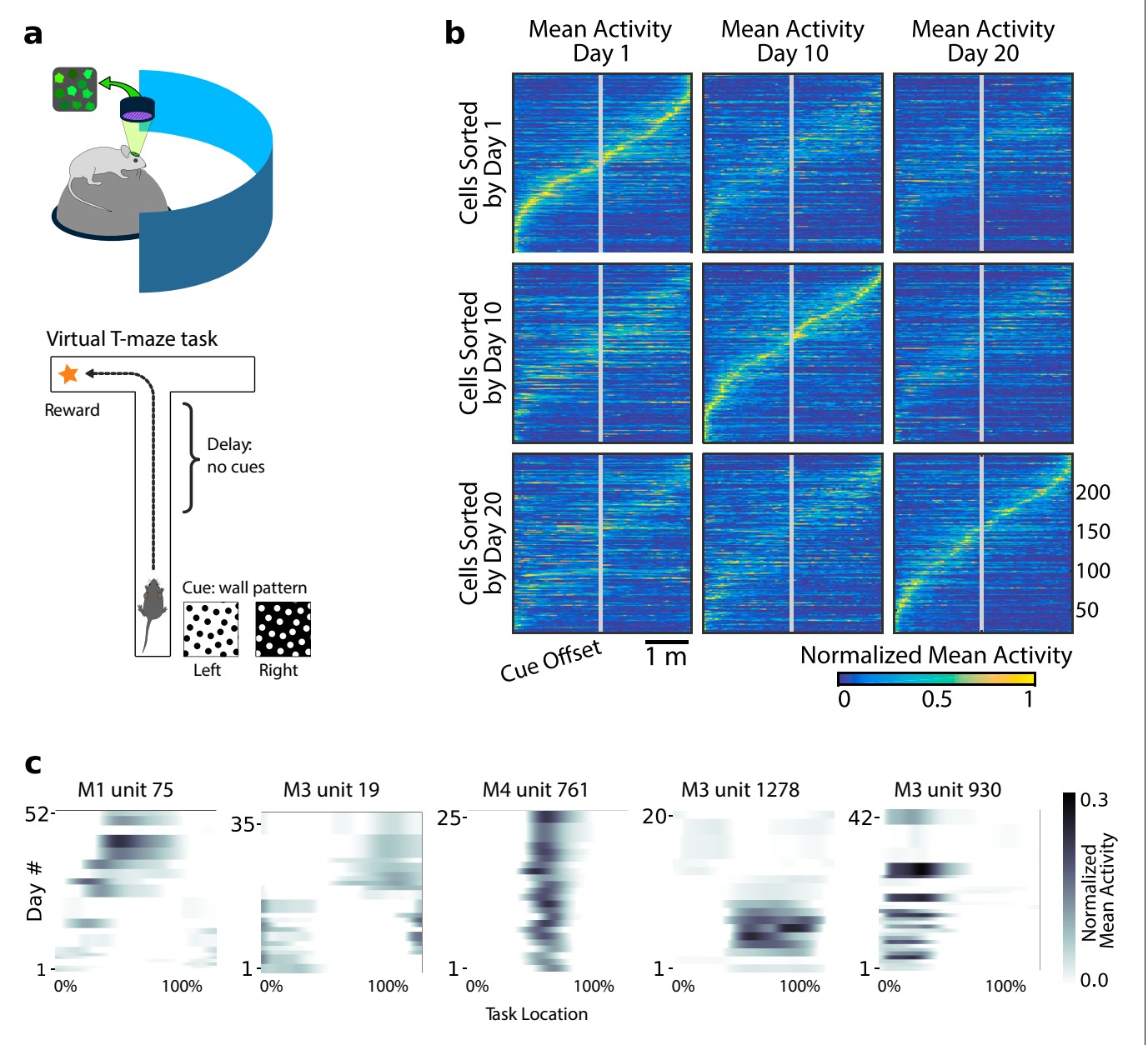

**Figure 1.** Neural population coding of spatial navigation reconfigures over time in a virtual-reality maze task. (**a**) Mice were trained to use visual cues to navigate to a reward in a virtual-reality maze; neural population activity was recorded using $Ca^{2+}$ imaging *Driscoll et al., 2017*. (**b**) (Reprinted from *Driscoll et al., 2017*) Neurons in PPC (vertical axes) fire at various regions in the maze (horizontal axes). Over days to weeks, individual neurons change their tuning, reconfiguring the population code. This occurs even at steady-state behavioral performance (after learning). (**c**) Each plot shows how location-averaged normalized activity changes for single cells over weeks. Missing days are interpolated to the nearest available sessions, and both left and right turns are combined. Neurons show diverse changes in tuning over days, including instability, relocation, long-term stability, gain/loss of selectivity, and intermittent responsiveness.

accuracy suggests that brain areas with sufficiently dense connectivity to PPC can extract this information via simple weighted sums.

The number of PPC neurons recorded is a subset of the total PPC population. To assess whether additional neurons might improve decoding accuracy, we evaluated decoding performance of randomly drawn subsets of recorded neurons (*Figure 2c*). Extrapolation of the decoding performance

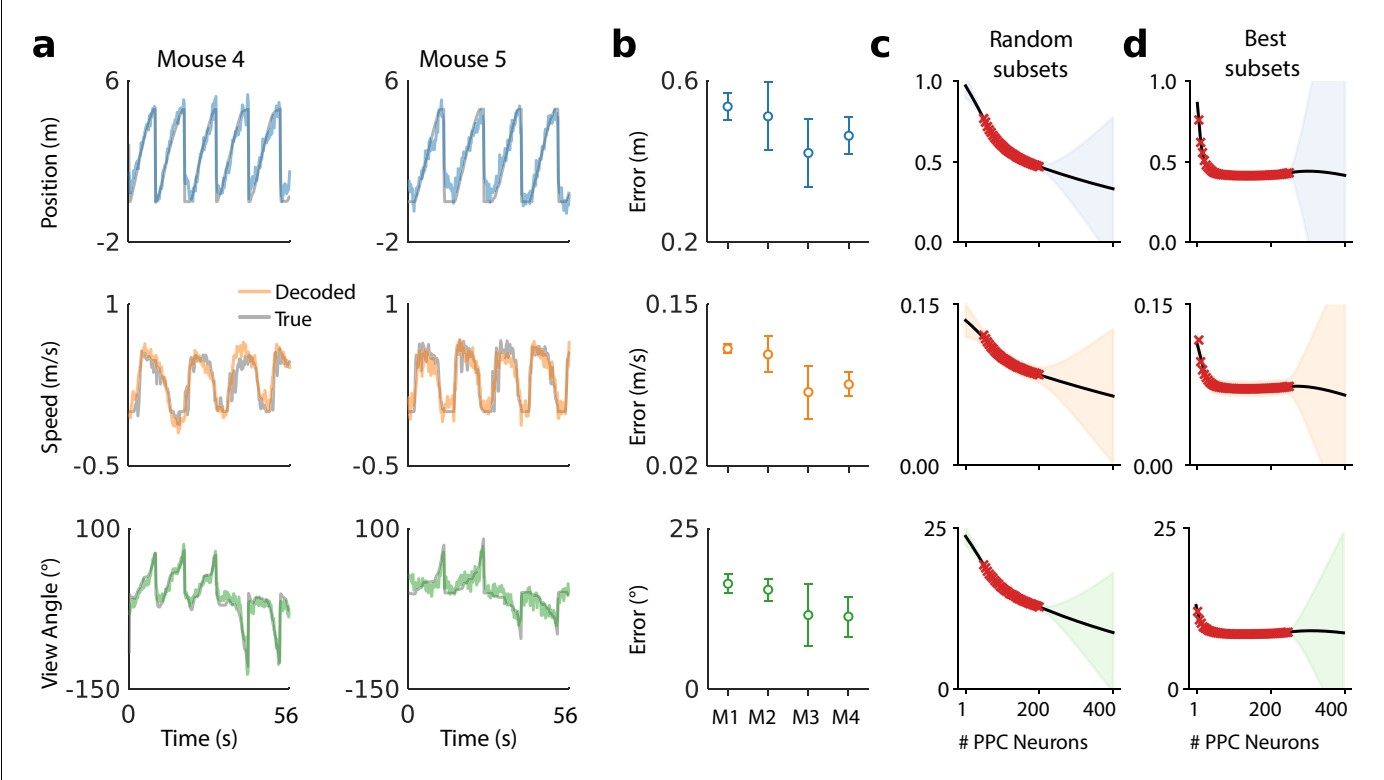

**Figure 2.** A linear decoder can extract kinematic information from PPC population activity on a single day. (**a**) Example decoding performance for a single session for mice 4 and 5. Grey denotes held-out test data; colors denote the prediction for the corresponding kinematic variable. (**b**) Summary of the decoding performance on single days; each point denotes one mouse. Error bars denote one standard deviation over all sessions that had at least $N=200$ high-confidence PPC neurons for each mouse. (Mouse two is excluded due to an insufficient number of isolated neurons). Chance level is ~1.5 m for forward position, and varies across subjects for forward velocity (~0.2–0.25 m/s) and head direction (~20-30 ). (**c**) Extrapolation of the performance of the static linear decoder for decoding position as a function of the number of PPC neurons, done via Gaussian process regression (Materials and methods). Red '×' marks denote data; solid black line denotes the inferred mean of the GP. Shaded regions reflect ±1.96σ Gaussian estimates of the 95th and 5th percentiles. (**d**) Same as panel (**c**), but where the neurons have been ranked such that the 'best' subset of size $1 \le K \le N$ is chosen, selected by greedy search based on explained variance (Materials and methods: Best K-Subset Ranking).

The online version of this article includes the following figure supplement(s) for figure 2:

**Figure supplement 1.** Behavioral stability.

suggested that better performance might be possible with a larger population of randomly sampled PPC neurons than we recorded.

It is possible that a random sample of neurons misses the 'best' subset of cells for decoding task variables. When we restricted to optimal subsets of neurons we found that performance improved rapidly up to ~30 neurons and saturated at ~30%(50–100 neurons) of the neurons recorded (*Figure 2d*). On a given day task variables could be decoded well with relatively few (~10) neurons. However, the identity of the neurons in this optimal subset changed over days. For all subjects, no more than 1% of cells were consistently ranked in the top 10%, an no more than 13% in the top 50%. We confirmed that this instability was not due to under-regularization in training (Materials and methods: Best K-Subset Ranking).

Of the neurons with strong location tuning, *Driscoll et al., 2017* found that 60% changed their location tuning over two weeks and a total of 80% changed over the 30- day period examined. We find that even the small remaining 'stable' subset of neurons exhibited daily variations in their Signal-to-Noise Ratio (SNR) with respect to task decoding, consistent with other studies (*Carmena et al., 2005*). For example, no more than 8% of neurons that were in the top 25% in terms of tuning-peak stability were also consistently in the top 25% in terms of SNR for all days. If a neuron becomes relatively less reliable, then the weight assigned may become inappropriate for decoding.

This affects our analyses, and would also physiologically affect a downstream neuron with fixed synaptic weights.

## Representational drift is systematic and significantly degrades a fixed readout

Naively fitting a linear model to data from any given day shows that behavioral variables are encoded in a way that permits a simple readout, but there is no guarantee that this readout will survive long-term drift in the neural code. To illustrate this, we compared the decoding performance of models fitted on a given day with decoders optimized on data from earlier or later days. We restricted this analysis to those neurons that were identified with high confidence on all days considered. We found that decoding performance decreased as the separation between days grew (*Figure 3a*). This is unsurprising given the extent of reconfiguration reported in the original study (*Driscoll et al., 2017*) and depicted in *Figure 1*. Furthermore, because task-related PPC activity is distributed over many neurons, many different linear decoders can achieve similar error rates due to the degeneracy in the representation (*Rokni et al., 2007*; *Kaufman et al., 2014*; *Montijn et al., 2016*). Since the directions in population activity used for inter-area communication might differ from the directions that maximally encode stimulus information in the local population (*Ni et al., 2018*; *Semedo et al., 2019*), single-day decoders might overlook a long-term stable subspace used for encoding and communication. This motivates the question of whether a drift-invariant linear decoder exists and whether its existence is biologically plausible.

To address this, we tested the performance of a single linear decoder optimized across data from multiple days. We concatenated data from different days using the same subset of PPC neurons (*Figure 3b*). In all four subjects, we found that such fixed multiple-day linear 'concatenated' decoders could recover accurate task variable information despite ongoing changes in PPC neuron tuning. However, the average performance of the multiple-day decoders was significantly worse than single-day linear decoders for each day (*Figure 3c*).

The existence of a fixed, approximate decoder implies a degenerate representation of task variables in the population activity of PPC neurons. In other words, there is a family of linear decoders that can recover behavioral variables while allowing weights to vary in some region of weight space. This situation is illustrated in *Figure 3b*, which depicts regions of good performance of single-day linear decoders as ellipsoids. The existence of an approximate concatenated decoder implies that these ellipsoids intersect over several days for some allowable level of error in the decoder. For a sufficiently redundant neural code, one might expect to find an invariant decoder for some specified level of accuracy even if the underlying code drifts. However, there are many qualitative ways in which drift can occur in a neural code: it could resemble a random walk, as some studies suggest (*Moczulska et al., 2013*; *Loewenstein et al., 2011*; *Loewenstein et al., 2015*), or there could be a systematic component. Is the accuracy we observe in the concatenated decoder expected for a random walk? In all subjects, we found that a concatenated decoder performed substantially better on experimental data than on randomly drifting synthetic data with matched sparseness and matched within/between-session variability (*Figure 3d*). This suggests that the drift in the neural data is not purely random.

We further investigated the dynamics of drift by quantifying the direction of changes in neural variability over time (*Figure 4c,d*, Materials and methods: Drift alignment). We found that drift is indeed aligned above chance to within-session neural population variability. This suggests that the biological mechanisms underlying drift are in part systematic and constrained by a requirement to keep a consistent population code over time. In comparison, the projection of drift onto behavior-coding directions was small, but still above chance. This is consistent with the hypothesis that ongoing compensation might be needed for a long-term stable readout.

To quantify the systematic nature of drift further, we modified the null model to make drift partially systematic by constraining the null-model drift within a low rank subspace (*Figure 4—figure supplement 1*). This reflects a scenario in which only a few components of the population code change over time. We found that the performance of a concatenated decoder for low-rank drift better approximated experimental data. For three of the four mice we could match concatenated decoder performance when the dimension of the drift process was constrained within a range of 14–26, a relatively small fraction (around 20%) of the components of the full population.

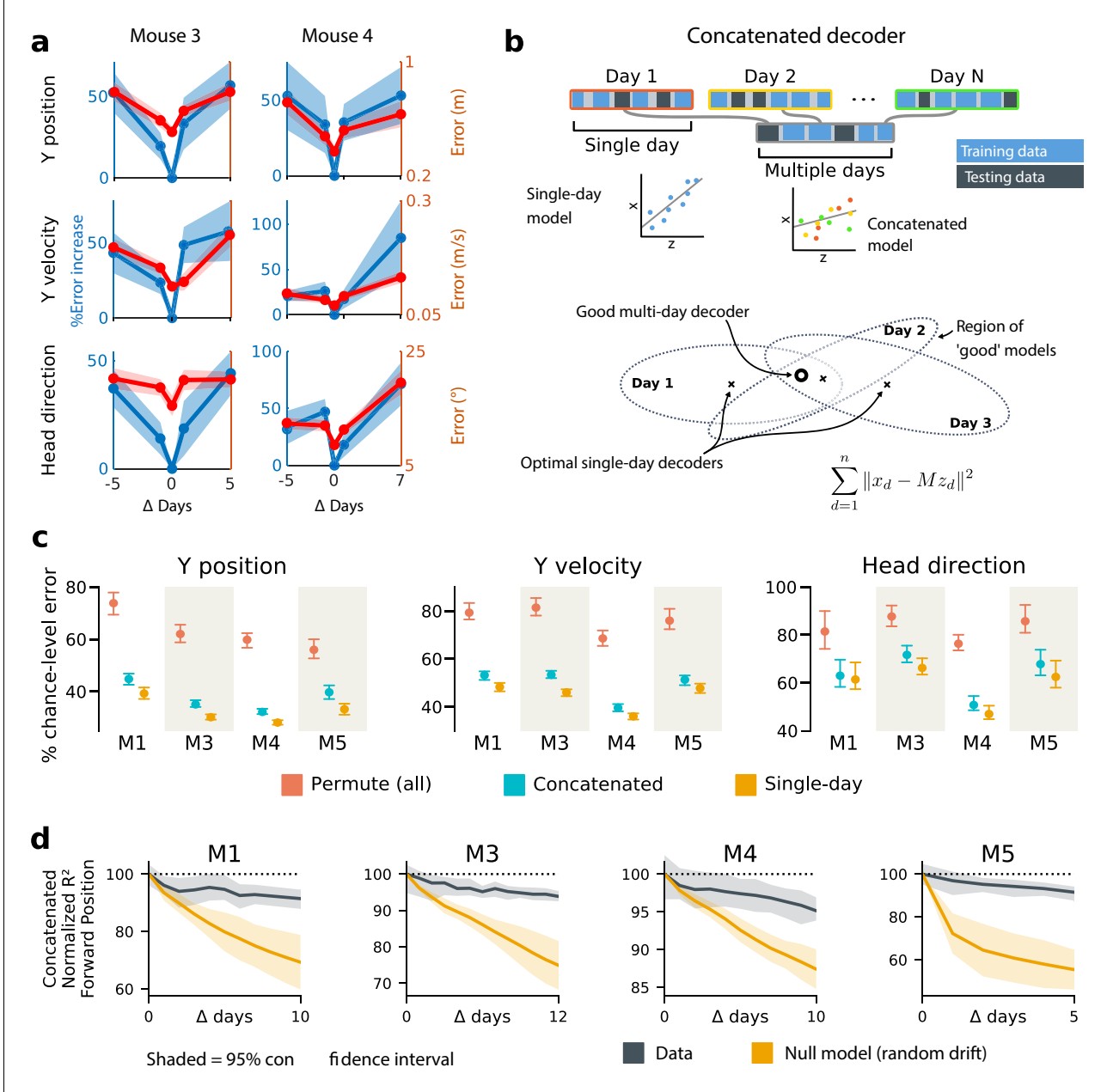

**Figure 3.** Single-day decoders generalize poorly to previous and subsequent days, but multi-day decoders exist with good performance. (a) Blue: % increase in error over the optimal decoder for the testing day (mouse 3, 136 neurons; mouse 4, 166 neurons). Red: Mean absolute error for decoders trained on a single day ('0') and tested on past/future days. (b) Fixed decoders $M$ for multiple days $d \in 1 \ldots D$ ('concatenated decoders') are fit to concatenated excerpts from several sessions. The inset equation reflects the objective function to be minimized (Methods). Due to redundancy in the neural code, many decoders can perform well on a single day. Although the single-day optimal decoders vary, a stable subspace with good performance can exist. (c) Concatenated decoders (cyan) perform slightly but significantly worse than single-day decoders (ochre; Mann-Whitney U test, p<0.01). They also perform better than expected if neural codes were unrelated across days (permutation tests; red). Plots show the mean absolute decoding error as a percent of the chance-level error (points: median, whiskers: 5th–95th%). Chance-level error was estimated by shuffling kinematics traces relative to neural time-series (mean of 100 samples). For the permutation tests, 100 random samples were drawn with the neuronal identities randomly permuted. (d) Plots show the rate at which concatenated-decoder accuracy (normalized $R^2$) degrades as the number of days increase. Concatenated decoders (black) degrade more slowly than expected for random drift (ochre). Shaded regions reflect the inner 95% of the data (generated by resampling for the null model). The null model statistics are matched to the within- and between-day variance and sparsity of the experimental data for each animal (Materials and methods).

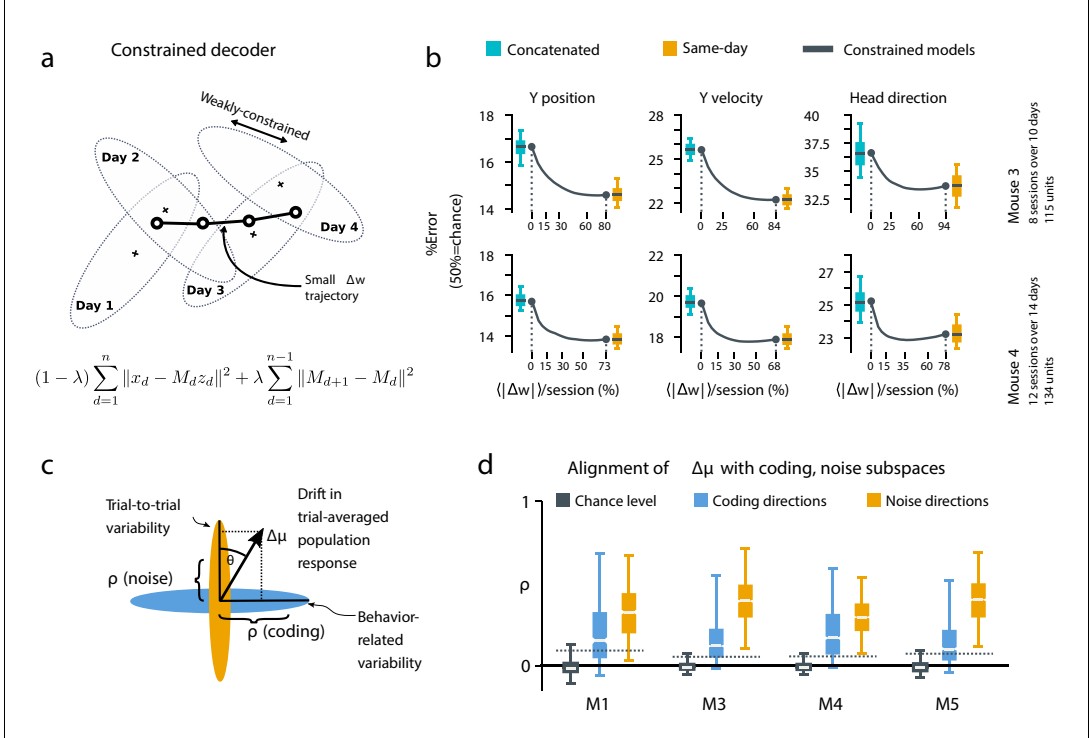

**Figure 4.** A slowly-varying component of drift disrupts the behavior-coding subspace. (**a**) The small error increase when training concatenated decoders (*Figure 3*) suggests that plasticity is needed to maintain good decoding in the long term. We assess the minimum rate for this plasticity by training a separate decoder $M_d$ for each day, while minimizing the change in weights across days. The parameter λ controls how strongly we constrain weight changes across days (the inset equation reflects the objective function to be minimized; Methods). (**b**) Decoders trained on all days (cyan) perform better than chance (red), but worse than single-day decoders (ochre). Black traces illustrate the plasticity-accuracy trade-off for adaptive decoding. Modest weight changes per day are sufficient to match the performance of single-day decoders (Boxes: inner 50% of data, horizontal lines: median, whiskers: 5–95th%). (**c**) Across days, the mean neural activity associated with a particular phase of the task changes (Δμ). We define an alignment measure ρ (Materials and methods) to assess the extent to which these changes align with behavior-coding directions in the population code (blue) verses directions of noise correlations (ochre). (**d**) Drift is more aligned with noise (ochre) than it is with behavior-coding directions (blue). Nevertheless, drift overlaps this behavior-coding subspace much more than chance (grey; dashed line: 95% Monte-Carlo sample). Each box reflects the distribution over all maze locations, with all consecutive pairs of sessions combined.

The online version of this article includes the following figure supplement(s) for figure 4:

**Figure supplement 1.** Concatenated decoder performance depends on the rank of the drift.

## Biologically achievable rates of plasticity can compensate drift, independent of specific learning rules

Together, these analyses show that the observed dynamics of drift favor a fixed linear readout above what would be expected for random drift. However, our results also show that a substantial component of drift cannot be confined to the null space of a fixed downstream linear readout. We asked how much ongoing weight change would be needed to achieve the performance of single-day decoders while minimizing day-to-day changes in decoding weights. We first approached this without assuming a specific plasticity rule, by simultaneously optimizing linear decoders for all recorded days while penalizing the magnitude of weight change between sessions (*Figure 4a*, Materials and methods: Concatenated and constrained analyses). By varying the magnitude of the weight change penalty we interpolated between the concatenated decoder (no weight changes) and the single-day decoders (optimal weights for each day). The result of this is shown in *Figure 4b*. Performance improves rapidly once small weight changes are permitted (~12–25% per session). Thus, relatively modest amounts of synaptic plasticity might suffice to keep encoding consistent with changes in representation, provided a mechanism exists to implement appropriate weight changes.

## A biologically plausible local learning rule can compensate drift

The results in *Figure 4b* suggest that modest amounts of synaptic plasticity could compensate for drift, but do not suggest a biologically plausible mechanism for this compensation. Could neurons track slow reconfiguration using locally available signals in practice? To test this, we used an adaptive linear neuron model based on the least mean square learning (LMS) rule (*Widrow and Hoff, 1960*; *Widrow and Hoff, 1962*) (Materials and methods). This algorithm is biologically plausible because it only requires each synapse to access its current weight and recent prediction error (*Figure 5a*, Materials and methods: Online LMS algorithm).

*Figure 5b* shows that this online learning rule achieved decoding performance comparable to the offline constrained decoders. Over the timespan of the data, LMS allows a linear decoder to track representational drift observed (*Figure 5c*), exhibiting weight changes of ~10%/day across all animals (learning rate $4 \times 10^{-4}$/sample, *Figure 5—figure supplement 1*). These results suggest that small weight changes could track representational drift in practice. In contrast, we found that LMS struggled to match the unconstrained drift of the null model explored in *Figure 3d*. Calibrating the LMS learning rate on the null model to match the mean performance seen on the true data required

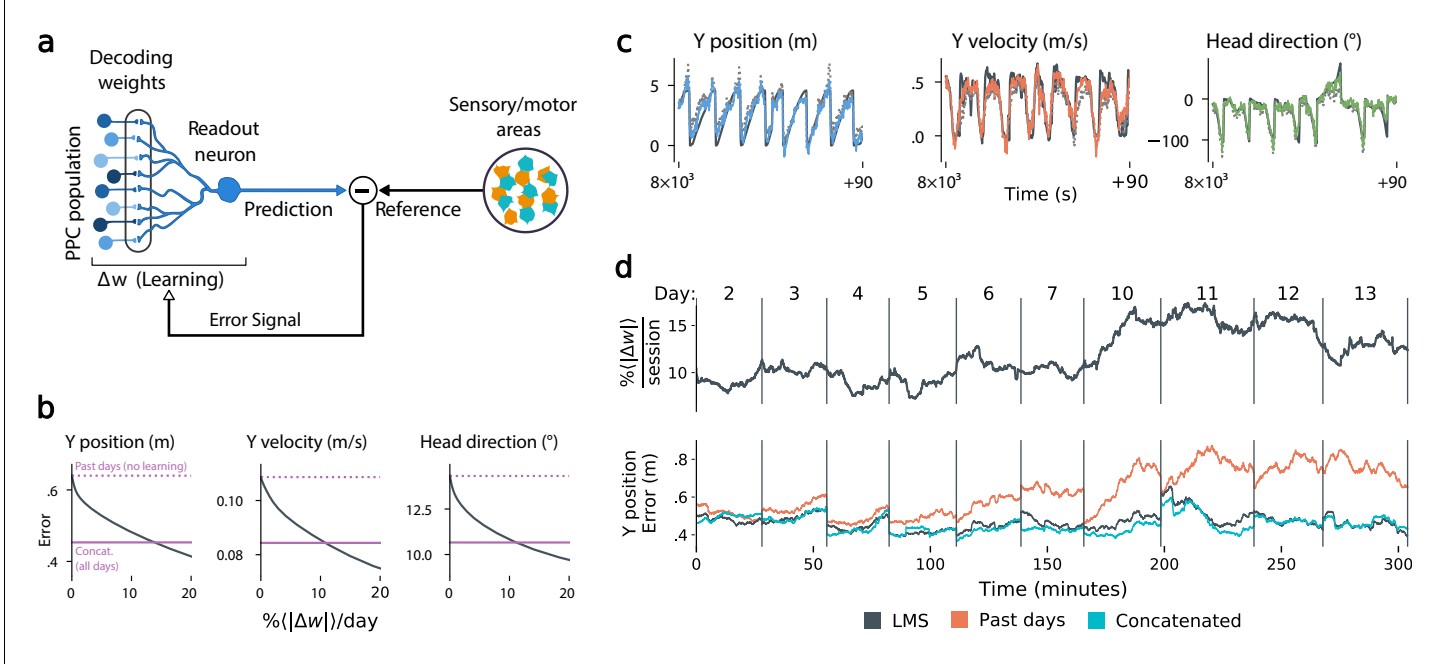

**Figure 5.** Local, adaptive decoders can track representational drift over multiple days. (a) The Least Mean-Squares (LMS) algorithm learns to linearly decode a target kinematic variable based on error feedback. Continued online learning can track gradual reconfiguration in population representations. (b) As the average weight change per day (horizontal axis) increases, the average decoding error (vertical axis) of the LMS algorithm improves, shown here for three kinematic variables (Mouse 4, 144 units, 10 sessions over 12 days; Methods: methods:lms). (Dashed line: error for a decoder trained on only the previous session without online learning; Solid line: performance of a decoder trained over all testing days). As the rate of synaptic plasticity is increased, LMS achieves error rates comparable to the concatenated decoder. (c) Example LMS decoding results for three kinematic variables. Ground truth is plotted in black, and LMS estimate in color. Sample traces are taken from day six. Dashed traces indicate the performance of the decoder without ongoing re-training. (d) (top) Average percent weight-change per session for online decoding of forward position (learning rate: $4 \times 10^{-4}$/sample). The horizontal axis reflects time, with vertical bars separating days. The average weight change is 10.2% per session. To visualize %$\Delta w$ continuously in this plot, we use a sliding difference with a window reflecting the average number of samples per session. (bottom) LMS (black) performs comparably to the concatenated decoder (cyan) (LMS mean absolute error of 0.47 m is within $\leq$ 3% of concatenated decoder error). Without ongoing learning, the performance of the initial decoder degrades (orange). Error traces have been averaged over ten minute intervals within each session. Discontinuities between days reflect day-to-day variability and suggest a small transient increase in error for LMS decoding at the start of each day.

The online version of this article includes the following figure supplement(s) for figure 5:

**Figure supplement 1.** Online learning with LMS: additional subjects.

**Figure supplement 2.** The plasticity level required to track drift varies with population size.

**Figure supplement 3.** Extrapolation to larger populations.

an average weight change of 93% per day. In comparison, matching the average percent weight change per day of 10%, the null model produced a normalized mean-squared-error of 1.3 $\sigma^2$ (averaged over all mice), worse than chance. This further indicates that drift is highly structured, facilitating online compensation with a local learning rule.

We stress that modeling assumptions mean that these results are necessarily a proxy for the rates of synaptic plasticity that are observed in vivo. Nonetheless, we believe these calculations are conservative. We were restricted to a sample of ~100–200 neurons, at least an order of magnitude less than the typical number of inputs to a pyramidal cell in cortex. The per-synapse magnitude of plasticity necessarily increases when smaller subsets are used for a readout (*Figure 5—figure supplement 2*). One would therefore expect lower rates of plasticity for larger populations. Indeed, when we combined neurons across mice into a large synthetic population (1238 cells), we found that the plasticity required to achieve target error asymptotes at less than 4% per day (*Figure 5—figure supplement 3*). Together, these results show a conservatively achievable bound on the rate of plasticity required to compensate drift in a biologically plausible model.

## Discussion

Several theories have been proposed for how stable behavior could be maintained despite ongoing changes in connectivity and neural activity. Here, we found that representational drift occurred in both coding and non-coding subspaces. On a timescale of a few days, redundancy in the neural population could accommodate a significant component of drift, assuming a biological mechanism exists for establishing appropriate readout weights. Simulations suggested that the existence of this approximately stable subspace were not simply a result of population redundancy, since random diffusive drift quickly degraded a downstream readout. Drift being confined to a low-dimensional subspace is one scenario that could give rise to this, although we do not exclude other possibilities. Nevertheless, a non-negligible component of drift resides outside the null space of a linear encoding subspace, implying that drift will eventually destroy any fixed-weight readout.

However, we showed that this destructive component of drift could be compensated with small and biologically realistic changes in synaptic weights, independently of any specific learning rule. Furthermore, we provided an example of a simple and biologically plausible learning rule that can achieve such compensation over long timescales with modest rates of plasticity. If our modeling results are taken literally, this would suggest that a single unit with connections to ~100 PPC neurons can accurately decode task information with modest changes in synaptic weights over many days. This provides a concrete and quantitative analysis of the implications of drift on synaptic plasticity and connectivity. Together, our findings provide some of the first evidence from experimental data that representational drift could be compatible with long-term memories of learned behavioral associations.

A natural question is whether a long-term stable subspace is supported by an unobserved subset of neurons that have stable tuning (*Clopath et al., 2017*). We do not exclude this possibility because we measured a subset of the neural population. However, over multiple samples from different animals our analyses consistently suggest that drift will reconfigure the code entirely over months. Specifically, we found that past reliability in single cells is no guarantee of future stability. This, combined with an abundance of highly-informative cells on a single day, contributes to poor (fixed) decoder generalization, because previously reliable cells eventually drop out or change their tuning. Consistent with this, studies have shown that connectivity in mammalian cortex is surprisingly dynamic. Connections between neurons change on a timescale of hours to days with a small number of stable connections (*Holtmaat et al., 2005*; *Minerbi et al., 2009*; *Holtmaat and Svoboda, 2009*; *Attardo et al., 2015*).

We stress that the kind of reconfiguration observed in PPC is not seen in all parts of the brain; primary sensory and motor cortices can show remarkable stability in neural representations over time (*Gallego et al., 2020*). However, even if stable representations exist elsewhere in the brain, PPC still must communicate with these areas. We suggest that a component of ongoing plasticity maintains congruent representations across different neural circuits. Such maintenance would be important in a distributed, adaptive system like the brain, in which multiple areas learn in parallel. How this is achieved is the subject of intense debate (*Rule et al., 2019*). We hypothesize that neural circuits have continual access to two kinds of error signals. One kind should reflect mismatch between

internal representations and external task variables, and another should reflect prediction mismatch between one neural circuit and another. Our study therefore motivates new experiments to search for neural correlates of error feedback between areas, and suggests further theoretical work to explore the consequences of such feedback.

## Materials and methods

### Data acquisition
The behavioral and two-photon calcium imaging data analyzed here were provided by the Harvey lab. Details regarding the experimental subjects and methods are provided in *Driscoll et al., 2017*.

### Virtual reality task
Details of the virtual reality environment, training protocol, and fixed association navigation task are described in *Driscoll et al., 2017*. In brief, virtual reality environments were constructed and operated using the MATLAB-based ViRMEn software (Virtual Reality Mouse Engine) *Harvey et al., 2012*. Data were obtained from mice that had completed the 4–8 week training program for the two-alternative forced choice T-maze task. The length of the virtual reality maze was fixed to have a total length of 4.5 m. The cues were patterns on the walls (black with white dots or white with black dots), and were followed by a gray striped 'cue recall' segment (2.25 m long) that was identical across trial types.

### Data preparation and pre-processing
Raw $Ca^{2+}$ fluorescence videos (sample rate=5.3Hz) were corrected for motion artefacts, and individual sources of $Ca^{2+}$ fluorescence were identified and extracted (*Driscoll et al., 2017*). Processed data consisted of normalized $Ca^{2+}$ fluorescence transients ('$\Delta F/F$') and behavioral variables (mouse position, view angle, and velocity). Inter-trial intervals (ITIs) were removed for all subsequent analyses. For offline decoding, we considered only correct trials, and all signals were centered to zero-mean on each trial as a pre-processing step.

When considering sequences of days, we restricted analysis to units that were continuously tracked over all days. For *Figures 3* and *4*, we use the following data: M1: seven sessions, 15 days, 101 neurons; M3: 10 sessions, 13 days, 114 neurons; M4: 10 sessions, 11 days, 146 neurons; M5: seven sessions, 7 days, 112 neurons. We allowed up to two-day recording gaps between consecutive sessions from the same mouse.

### Quantification and statistical analysis
#### Decoding analyses
We decoded kinematics time-series $\mathbf{x}=\{x_1, ..., x_T\}$ with $T$ time-points from the vector of instantaneous neural population activity $\mathbf{z}=\{z_1, ..., z_T\}$, using a linear decoder with a fixed set of weights $M$, that is $\hat{\mathbf{x}} = M^{\top}\mathbf{z}$. We used the ordinary least-squares (OLS) solution for $M$, which minimizes the squared (L2) prediction error $\varepsilon=\|\mathbf{x}-M^{\top}\mathbf{z}\|^2$ over all time-points. For the 'same-day' analyses, we optimize a separate $M_d$ for each day $d$ (*Figure 2*), restricting analysis to sessions with at least 200 identified units. We assessed decoding performance using 10-fold cross-validation, and report the mean absolute error, defined as $\langle\,|\,\mathbf{x}-\hat{\mathbf{x}}\,|\,\rangle$. Here, $|\,.\,|$ denotes the element-wise absolute value, and $\langle.\rangle$ denotes expectation.

### Best K-Subset ranking
For *Figure 2d*, we ranked cells in order of explained variance using a greedy algorithm. Starting with the most predictive cell, we iteratively added the next cell that minimized the MSE under ten-fold cross-validated linear decoding. To accelerate this procedure, we pre-computed the mean and covariance structure for training and testing datasets. MSE fits and decoding performance can be computed directly from these summary statistics, accelerating the several thousand evaluations required for greedy selection. We added L2 regularization to this analysis by adding a constant $\lambda I$ to the covariance matrix of the neural data. The optimal regularization strength ($\lambda = 10^{-4}$ to $10^{-3}$) slightly reduced decoding error, but did not alter the ranking of cells.

## Extrapolation via GP regression

To qualitatively assess whether decoding performance saturates with the available number of recorded neurons, we computed decoding performance on a sequence of random subsets of the population of various sizes (*Figure 2c,d*). Results for all analyses are reported as the mean over 20 randomly-drawn neuronal sub-populations, and over all sessions that had at least $N=150$ units. Gaussian process (GP) regression was implemented in Python, using a combination of a Matérn kernel and an additive white noise kernel. Kernel parameters were optimized via maximum likelihood (Scikit-learn, *Pedregosa et al., 2011*).

## Concatenated and constrained analyses

For both the concatenated (*Figure 3b,e*) and constrained analyses (*Figure 4a,b*), we used the set of identified neurons included in all sessions considered. In the concatenated analyses, we solved for a single decoder $M_c$ for all days:

$$\varepsilon = \sum_{d=1}^{n} \|\mathbf{x}_d - M_c^\top \mathbf{z}_d\|^2, \tag{1}$$

where $\varepsilon$ denotes the quadratic objective function to be minimized. In the constrained analysis, we optimized a series of different weights $\mathbf{M}=\{M_1,...,M_D\}$ for each day $d\in 1...D$, and added an adjustable L2 penalty $\lambda$ on the change in weights across days. This problem reduces to the 'same-day' analysis for $\lambda=0$, and approaches the concatenated decoder as $\lambda$ approaches 1:

$$\varepsilon = (1-\lambda) \sum_{d=1}^{n} \|\mathbf{x}_d - M_d^\top \mathbf{z}_d\|^2 + \lambda \sum_{d=1}^{n-1} \|M_{d+1} - M_d\|^2. \tag{2}$$

For the purposes of the constrained analysis, missing days were ignored and the remaining days treated as if they were contiguous. Two sessions were missing from the 10 and 14 day spans for mice 3 and 4, respectively (*Figure 4b*). *Figure 3c* also shows the expected performance of a concatenated decoder for completely unrelated neural codes. To assess this, we permuted neuronal identities within individual sessions, so that each day uses a different "code'.

## Null model

We developed a null model to assess whether the performance of the concatenated decoder was consistent with random drift. For this, we matched the amount of day-to-day drift based on the rate at which single-day decoders degrade. We also sampled neural states from the true data to preserve sparsity and correlation statistics. The null model related neural activity to a 'fake' observable readout (e.g. mouse position) via an arbitrary linear mapping. The null model changed from day to day, reflecting drift in the neural code. The fidelity of single day and across day decoders in inferring a readout from the null model was matched to the true data.

For each animal, we take the matrix $z \in \mathbb{R}^{n \times d}$ of mean-centered neural activity on day one, where $n$ represents the number of recorded neurons and $d$ represents the number of datapoints. We relate this matrix to pseudo-observations of mouse position $z$ via a null model of the form $z_r = M_r^\top z + \epsilon_r$, where $M_r^\top, \epsilon_r \in \mathbb{R}^{1 \times n}$. Note that $r$ indexes days. The vector $\epsilon_r$ is generated as scaled i.i.d. Gaussian noise. We scale $\epsilon_r$ such that the accuracy of a linear decoder trained on the data $(z, x_r)$ matches the average (over days) accuracy of a single-day decoder trained on the true data.

Next, we consider the choice of the randomly-drifting readout, $M_r$. On day one, $M_1$ is generated as a vector of uniform random variables on $[0, 1]$. Given $M_r$, we desire an $M_{r+1}$ that satisfies.

- $\|M_{r+1}\|_2 = \|M_r\|_2$.
- The expected coefficient of multiple correlation of $x_{r+1} = M_{r+1}^\top z$ against the predictive model $M_r^\top z$ (between day $R^2$) matches the average (over days) of the equivalent statistic generated from the true data.

To do this, we first generate a candidate $\Delta M_r' \in \mathbb{R}^{n \times 1}$ as a vector of i.i.d. white noise. The components of $\Delta M_r'$ orthogonal and parallel to $M_r$ are then scaled so that $M_{r+1} = M_r + \Delta M_r$ satisfies the constraints above.

In *Figure 4—figure supplement 1, a* modification of the null model that confined inter-day model drift to a predefined subspace was used. Before simulating the null model over days, we randomly chose $k$ orthogonal basis vectors, representing a $k$-dimensional subspace. We then searched for a candidate $\Delta M'_r$, on each inter-day interval, that was representable as a weighted sum of these basis vectors. This requirement was in addition to those previously posed. Finding such a $\Delta M'_r$ corresponds to solving a quadratically-constrained quadratic program. This is non-convex, and thus a solution will not necessarily be found. However, solutions were always found in practice. We used unit Gaussian random variables as our initial guesses for each component of $\Delta M'_r$, before solving the quadratic program using the IPOPT toolbox (*Wächter and Biegler, 2006*).

## Drift alignment

We examine how much drift aligns with noise correlations verses directions of neural activity that vary with the task ('behavior-coding directions'). We define an alignment statistic $\rho$ that reflects how much drift projects onto a given subspace (i.e. noise vs. behavior). We normalize $\rho$ so that 0 reflects chance-level alignment and one reflects perfect alignment of the drift with the largest eigenvector of a given subspace (e.g. the principal eigenvector of the noise covariance).

Let $z(x)$ denote the neural population activity, where $x$ reflects a normalized measure of maze location, akin to trial pseudotime. Define drift $\Delta \mu_z(x)$ as the change in the mean neural activity $\mu_z(x)$ across days. We examine how much drift aligns with noise correlations verses directions of neural activity that vary with task pseudotime ($dz(x)/dx$).

To measure the alignment of a drift vector $\Delta \mu$ with the distribution of inter-trial variability (i.e. noise), we consider the trial-averaged mean $\mu$ and covariance $\Sigma$ of the neural activity (log calcium-fluorescence signals filtered between 0.03 and .3 Hz and z-scored), conditioned on trial location and the current/previous cue direction. We use the mean squared magnitude of the dot product between the change in trial-conditioned means between days ($\Delta \mu$), with the directions of inter-trial variability ($\Delta z = z - <z>$) on the first day, which is summarized by the product $\Delta \mu^\top \Sigma \Delta \mu$:

$$
\begin{aligned}
\left\langle |\Delta \mu^\top \Delta z|^2 \right\rangle &= \left\langle \Delta \mu^\top \Delta z \Delta z^\top \Delta \mu \right\rangle \\
&= \Delta \mu^\top \left\langle \Delta z \Delta z^\top \right\rangle \Delta \mu \\
&= \Delta \mu^\top \Sigma \Delta \mu.
\end{aligned}
\tag{3}
$$

To compare pairs of sessions with different amounts of drift and variability, we normalize the drift vector to unit length, and normalize the trial-conditioned covariance by its largest eigenvalue $\lambda_{\max}$:

$$
\phi_{\text{trial}}^2 = \frac{\Delta \mu^\top \Sigma \Delta \mu}{|\Delta \mu|^2 \cdot \lambda_{\max}}
\tag{4}
$$

The statistic $\phi_{\text{trial}}$ equals one if the drift aligns perfectly with the direction of largest inter-trial variability, and can be interpreted as the fraction of drift explained by the directions of noise correlations.

Random drift can still align with some directions by chance, and the mean squared dot-product between two randomly-oriented $D$-dimensional unit vectors scales as $1/D$. Accounting for the contribution from each dimension of $\Sigma$, the expected chance alignment is therefore $\phi_0^2 = tr(\Sigma)/(D \cdot \lambda_{\max})$. We normalize the alignment coefficient $\rho_{\text{noise}}$ such that it is 0 for randomly oriented vectors, and one if the drift aligns perfectly with the direction of largest variability:

$$
\rho_{\text{noise}} = \frac{\phi_{\text{trial}} - \phi_0}{1 - \phi_0}
\tag{5}
$$

We define a similar alignment statistic $\rho_{\text{coding}}$ to assess how drift aligns with directions of neural variability that encode location. We consider the root-mean-squared dot product between the drift $\Delta \mu$, and the directions of neural activity ($z$) that vary with location ($x$) on a given trial, that is $\nabla_x z(x)$:

$$\begin{aligned}
\left\langle \Delta\mu^{\top}\nabla_x z(x)|^2 \right\rangle &= \left\langle \Delta\mu^{\top}[\nabla_x z(x)][\nabla_x z(x)]^{\top}\Delta\mu \right\rangle \\
&= \Delta\mu^{\top}\left\langle [\nabla_x z(x)][\nabla_x z(x)]^{\top}\right\rangle\Delta\mu \\
&= \Delta\mu^{\top}\left[\Sigma_{\nabla}+\mu_{\nabla}\mu_{\nabla}^{\top}\right]\Delta\mu
\end{aligned} \tag{6}$$

In contrast to the trial-to-trial variability statistic, this statistic depends on the second moment $\Sigma_{\nabla}+\mu_{\nabla}\mu_{\nabla}^{\top}$, where $\nabla_x z(x) \sim \mathcal{N}(\mu_{\nabla},\Sigma_{\nabla})$. We define a normalized $\phi^2_{\text{coding}}$ and $\rho_{\text{coding}}$ similarly to $\phi^2_{\text{trial}}$ and $\rho_{\text{noise}}$. For the alignment of drift with behavior, we observed $\rho_{\text{coding}}= 0.11–0.24$ ($\mu=0.15$, $\sigma=0.03$), which was significantly above chance for all mice. In contrast, the 95[th] percentile for chance alignment (i.e. random drift) ranged from 0.06 to 0.10 ($\mu=0.07$, $\sigma=0.02$). Drift aligned substantially more with noise correlations, with $\rho=0.29–0.43$ ($\mu=0.36$, $\sigma=0.04$).

## Online LMS algorithm

The Least Mean-Squares (LMS) algorithm is an online approach to training and updating a linear decoder, and corresponds to stochastic gradient-descent (*Figure 4a*). The algorithm was originally introduced in *Widrow and Hoff, 1960*; *Widrow and Hoff, 1962*; *Widrow and Stearns, 1985*. Briefly, LMS computes a prediction error for an affine decoder (i.e. a linear decoder with a constant offset feature or bias parameter) at every time-point, which is then used to update the decoding weights. We analyzed twelve contiguous sessions from mouse 4 (144 units in common), and initialized the decoder by training on the first two sessions using OLS.

By varying the learning rate, we obtained a trade-off (*Figure 4b*) between the rate of weight changes and the decoding error, with the most rapid learning rates exceeding the performance of offline (static) decoders. In *Figure 4d*, we selected an example with a learning rate of $\eta=4\times10^{-4}$. To provide a continuous visualization of the rate of weight change in *Figure 4d*, we used a sliding difference with a duration matching the average session length. This was normalized by the average weight magnitude to report percent weight change per day. In all other statistics, per-day weight change is assessed as the difference in weights at the end of each session, divided by the days between the sessions.

## Data and code availability

Datasets recorded in *Driscoll et al., 2017* are available from the Dryad repository (https://doi.org/10.5061/dryad.gqnk98sjq). The analysis code generated during this study is available on Github (https://github.com/michaelerule/stable-task-information; copy archived at https://github.com/elifesciences-publications/stable-task-information; *Rule, 2020*).

## Acknowledgements

We thank Fulvio Forni, Yaniv Ziv and Alon Rubin for in depth discussions. This work was supported by the Human Frontier Science Program (RGY0069), ERC Starting Grant (StG FLEXNEURO 716643) and grants from the NIH (NS089521, MH107620, NS108410)

## Additional information

### Funding

| Funder | Grant reference number | Author |
| --- | --- | --- |
| Human Frontier Science Program | RGY0069 | Michael E Rule<br>Adrianna R Loback<br>Christopher D Harvey<br>Timothy O'Leary |
| H2020 European Research Council | FLEXNEURO 716643 | Dhruva Raman<br>Timothy O'Leary |
| National Institutes of Health | NS089521 | Christopher D Harvey |
| National Institutes of Health | MH107620 | Christopher D Harvey |

| National Institutes of Health | NS108410 | Christopher D Harvey |

The funders had no role in study design, data collection and interpretation, or the decision to submit the work for publication.

## Author contributions

Michael E Rule, Conceptualization, Formal analysis, Validation, Investigation, Visualization, Methodology; Adrianna R Loback, Formal analysis, Investigation, Visualization, Methodology; Dhruva V Raman, Conceptualization, Validation, Investigation, Methodology; Laura N Driscoll, Data curation; Christopher D Harvey, Data curation, Funding acquisition, Project administration; Timothy O'Leary, Conceptualization, Supervision, Funding acquisition, Validation, Investigation, Visualization, Methodology, Project administration

## Author ORCIDs

Michael E Rule (iD) https://orcid.org/0000-0002-4196-774X
Timothy O'Leary (iD) https://orcid.org/0000-0002-1029-0158

## Decision letter and Author response

Decision letter https://doi.org/10.7554/eLife.51121.sa1
Author response https://doi.org/10.7554/eLife.51121.sa2

# Additional files

## Supplementary files

• Transparent reporting form

## Data availability

Datasets recorded in Driscoll et al., 2017, are available from the Dryad repository under the https://doi.org/10.5061/dryad.gqnk98sjq. The analysis code generated during this study is available on Github https://github.com/michaelerule/stable-task-information (copy archived at https://github.com/elifesciences-publications/stable-task-information).

The following dataset was generated:

| Author(s) | Year | Dataset title | Dataset URL | Database and Identifier |
|---|---|---|---|---|
| Driscoll LN | 2020 | Data from: Stable task information from an unstable neural population | https://doi.org/10.5061/dryad.gqnk98sjq | Dryad Digital Repository, 10.5061/dryad.gqnk98sjq |

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
