## [Decision Letter]

**Acceptance summary:**

This work addresses how accurate readout in the brain can be maintained despite shifts in neural population tuning and variability. The work reanalyzes previous data from posterior parietal cortex and digs deeper to show that a simple linear readout can, in fact, recover kinematic variables like animal position and speed from this drifting population. While this simple readout works well, it does slowly degrade over days. This work also shows how to ameliorate this degradation: plasticity that operates via a biologically plausible mechanism can maintain accurate readout.

**Decision letter after peer review:**

[Editors’ note: the authors submitted for reconsideration following the decision after peer review. What follows is the decision letter after the first round of review.]

Thank you for submitting your work entitled "Stable task information from an unstable neural population" for consideration by *eLife*. Your article has been reviewed by three peer reviewers, one of whom is a member of our Board of Reviewing Editors, and the evaluation has been overseen by a Senior Editor. The reviewers have opted to remain anonymous.

Our decision has been reached after consultation between the reviewers. Based on these discussions and the individual reviews below, we regret to inform you that your work will not be considered further for publication in *eLife*.

This is a clearly-presented initial study on how stable readout across days might be achieved despite shifting neural representations. The results have been judged to be sound, analytically, but the potential impact of the work falls short of threshold for a Short Report. Individual reviewer comments are listed below, but the main critiques are summarized here:

1) There are only data on 1 mouse to support the key result, which itself is not surprising given previous work from Driscoll et al., 2017.

2) The present work lacks a null model against which one can properly interpret the success of the concatenated decoder.

*Reviewer #1:*

This is a well-written, short manuscript about changes in neuronal activity patterns in PPC over days, and how stable readout can be achieved with a simple, linear decoder despite these shifting sands. The idea is that a single, best compromise, linear decoder can be found that is immune to the reconfiguration in the neural population. The work posits, but doesn't prove that the reconfiguration exists in the "null space" of the task.

There are a number of theoretical papers (as nicely referenced in this document) about how accurate decoders might be maintained in changing neural populations, but the upside of this work is that:

a) The results are taken from experimental data with large enough N's and over enough days that decoding accuracy can be traced, and

b) This is the simplest of all possible theories of how performance is maintained, and it's reasonably plausible.

I have some substantive concerns :

1) Given that the consensus decoder had to perform better across days than any single day decoder, it's not clear how surprising these results are.

2) It wasn't clear how well this extrapolates across different mice. In some figures, 3 or 4 individuals are compared, others just 2, others yet, just 1 mouse (mouse 4) is mentioned. This is central to the generality of the paper and should be laid out more clearly. Do the concatenated decoder and LMS decoder results hold for more than one individual?

3) The arguments about the scaling of the biologically plausible weight adjustments seem a little problematic. It's not clear why the results here form an upper bound on the weight changes needed to maintain accurate decoding. Also, it wasn't clear how the interactions between networks, maintaining congruence, is achieved. That final part of the Discussion was a bit vague.

*Reviewer #2:*

Loback et al. re-analyze data from Driscoll et al., 2017, which had previously shown that PPC representations are unstable over days during a delayed VR T-maze task. Here, using linear decoders, they find that a static decoder can do a reasonable job if trained on data from all days, and that an old model of synaptic weight updates can be applied to maintain decoder performance. The analyses seem to have been done reasonably, but the results strike me as rather shallow and are based on limited data.

The first main result is simply that unstable representations cause single-day (linear) decoders to generalize poorly, but a multi-day decoder to perform somewhat better. I have two issues with this result:

1) Given that activity is sparse and does not have systematic shifts in tuning, this decoding result is very nearly a mathematical necessity. Because of sparsity, the decoder likely ends up built so that different units drive the decoder performance on different sessions. This would not be news. Further, there are no null models analyzed for what would happen under different patterns of tuning shifts, which would be a helpful comparison. It is therefore not clear whether there is anything to be surprised at here.

2) How stable is the behavior within and across sessions? Details of behavior matter all over the brain (Stringer, Pachitariu et al., 2019, Musall, Kaufman et al., 2019), so it is possible that drift in behavioral details could lead to these shifts as well. At least, it should really be shown whether the parameters the authors track are stable over time.

These points said, there is value in this section. The quantifications of instability and of how many neurons are required for good decoder quality are helpful, and the point that only 6% of neurons are even in the top 50% of informativeness is surprising and interesting.

The second major result is the application of the Widrow and Hoff, 1962 model. If I understand correctly, this is primarily a different way of quantifying how fast the tuning changes occur (requiring ~2%/minute weight changes), and secondarily a proof of principle for that model. However, unless I'm missing something, this is a one-mouse result. That would not meet the standard for the field. In addition, comment #2 above applies to this result as well, making it harder to interpret.

*Reviewer #3:*

In their report, Loback and colleagues reanalyze data from Driscoll et al., 2017. They confirm the finding of that paper, namely that neuronal representations in the parietal cortex of mice reorganize over the time scale of days, while the overall information content is preserved. The authors then more specifically study the dynamics of these changes and relate them to simplified synaptic plasticity rules.

Overall, I find that the paper is clearly written and everything seems technically correct. However, I also find that it lacks scientific novelty. While I find the idea of linking the observed reorganization of neural activity with synaptic plasticity exciting, I find that the paper does not quite achieve that. I think the authors would need to work out some concrete consequences/constraints on plasticity for

this paper to become viable.

Broadly speaking, the current study is divided into two parts. The first part is a re-analysis of the data of Driscoll et al., 2017, which is performed in Figures 1-3. The authors use decoding methods to retrieve task information from the population activity. While some of the details of the population decoding methods are different to those used by the Driscoll et al., the overall conclusions are the same. The strongest point of the re-analysis is that the authors more clearly quantify the strength of the day-to-day changes using decoders that are constrained to change only little over days. That is a nice twist that was not performed in the Driscoll paper.

The second part of the paper is an attempt to relate these day-to-day changes to synaptic plasticity (Figures 3, 4). This part is rather brief and quite sketchy. Roughly, the authors simply reformulate the constrained decoder as an adaptive decoder. Conceptually, that is similar to the ideas brought forward by Rokni et al., Ajemian et al., and others. What could make this part interesting, is if this link could be made stronger, i.e., if it could really be a link to synaptic plasticity, rather than a link to a hypothetical readout. But even if the authors limit themselves to a single readout neuron, many questions are left unaddressed, e.g. how to extrapolate the adaptation rules for the decoder to realistic network sizes.

Other comments:

1) It was not clear to me what happens with the decoders within a session and between days. Do decoders 'jump' between days or stay roughly the same? How does that influence the adaptation rules?

2) Legend of Figure 4 and subsection “Biologically plausible weight adjustment can compensate for ongoing reconfiguration of PPC activity”. You repeatedly state that you approach the 'concatenated decoder.' I guess that should be the 'constrained decoder', otherwise it makes no sense to me.

[Editors’ note: further revisions were suggested prior to acceptance, as described below.]

Thank you for choosing to send your work entitled "Stable task information from an unstable neural population" for consideration at *eLife*. Your letter of appeal has been considered by a Senior Editor and a Reviewing editor, and we are prepared to consider a revised submission with no guarantees of acceptance.

Please address the following concerns that were raised in the discussion of your appeal and revised manuscript:

The null model added is a very nice one (Figure 3—figure supplement 2). It seems there has been a good effort to match it to properties of the data while incorporating a random walk. This is a crucial control. In addition, the new analysis of how drift aligns with coding vs. noise vs. chance (Figure 3—figure supplement 3) is also of substantial interest. Both of these new results are for 4 mice, which is excellent. The framing in the new manuscript also makes it somewhat clearer what the point of this paper is.

Points left to address in full:

1) Please go back and consider the more interesting null model in the other analyses and quantifications in this manuscript. This will improve many other parts of the paper. Please also place this new null model result in the main text of the paper.

2) Regarding the new null model:

Past evidence has clearly shown that neural tuning (or population activity) changes both randomly (assumed to be due to plasticity noise) and directionally (assumed to be due to feedback and learning). With the new null model, the analyses attempts to rule out a random walk. This is valuable effort. However, please add commentary on how this null model is useful despite ignoring the influence of the systematic, directional changes which were already demonstrated in the past, including the authors' own data, and which have usually been related to ongoing learning.

3) Please address in full the expanded review comments sent during the initial appeal. That text is reproduced here:

Thank you for sending us your thoughts and questions about the reviewer comments. This is an excellent piece of work, and the rejection is in no way about whether or not this is solid and publishable. The debate amongst the reviewers revolved around whether it was a significant enough advance for *eLife*. I have consulted with the reviewers in question and have a more thorough explanation of their comments. Please feel free to reach out if you have further questions.

This manuscript does, indeed, have some basic controls / null models. The shuffle control shows that the decoding is better than chance, and the static same-day model gives an idea of how much the weights have to change per session to do as well as freshly retrained decoders. The null models we'd like to see would compare results with more specific models. This is explained below:

The issue that I think all of the reviewers had is that it wasn't clear how much we should be surprised by these results, and we weren't clear on what new beliefs we should have after reading this paper if we've already read Driscoll, 2017.

There are two basic results that are claimed to be original. First: we should be surprised by the success of a concatenated decoder. On reviewer commented:

"Given that activity is sparse and does not have systematic shifts in tuning, this decoding result is very nearly a mathematical necessity. Because of sparsity, the decoder likely ends up built so that different units drive the decoder performance on different sessions. This would not be news."

In other words, to believe that there's something novel here, we would want to see a null model that can recapitulate the changes seen in Driscoll, 2017, with similar sparsity in the responses, where there *isn't* an ability to obtain a good concatenated decoder. We'd like to see what's required to have a different result. Without that, we would have expected that the concatenated decoder would work well.

Second, as it was understood by the reviewers, the manuscript argues that we should be surprised that updating decoder weights with the Widrow and Hoff model works here. From Driscoll, 2017, we have an idea of how rapidly location selectivity changes, and how rapidly a static decoder decays. Given that we know this, how rapidly would you expect to have to change the decoder? We didn't see much in the paper that wasn't just a different way of quantifying the same tuning changes. One reviewer suggested adding more specific null models because they think this would let the authors answer these deeper questions. For example, are all of the neurons smoothly changing their tuning? Do some change fast and others slow, and is this a continuous distribution? Is there coordination between neurons' tuning changes or are neurons changing independently? The current null models are extremes: the shuffle is related to a model where everything changes instantly (obviously wrong), and the same-day decoder is equivalent to there being no changes ever (which we know is wrong from Driscoll, 2017). So, what new have we learned?

Finally, regarding the result that only 6% of neurons are in the top 50% all 10 days: again, we lack the context to know how surprised we should be. If we suppose that the informative neurons are chosen randomly each day, then we'd expect the number of neurons that are in the top 50% for 10 days to be 0.5 ^ 9 = ~0.2%. In that case, 6% is surprisingly high. Looking at Driscoll's Figure 2B, ~40% of neurons keep their place preference for 10 days. In that case, 6% is surprisingly low. In fact, why is it so low? Could this just mean the decoder is under-regularized?

---

## [Author Response]

Editors’ note: The authors appealed the original decision. What follows is the authors’ response to the first round of review.]

This is a clearly-presented initial study on how stable readout across days might be achieved despite shifting neural representations. The results have been judged to be sound, analytically, but the potential impact of the work falls short of threshold for a Short Report. Individual reviewer comments are listed below, but the main critiques are summarized here:1) There are only data on 1 mouse to support the key result, which itself is not surprising given previous work from Driscoll et al., 2017.2) The present work lacks a null model against which one can properly interpret the success of the concatenated decoder.

We have completely addressed points (1) and (2) by extending the analysis across animals and by providing a null model for the concatenated decoder. We discuss details below. The outcome strengthens our conclusions. This, along with extensive additional analysis and rewriting to address remaining reviewers' comments means that the manuscript is significantly improved.

There was broad agreement between reviewers that the study (as previously presented) lacked depth and the importance of the results was not clear. Our original goal was to provide a short, sharp analysis with easily digestible results. We concede that in trying to keep the presentation terse we were too glib and superficial.

We have performed extensive additional analyses that strengthen our results. We have also rewritten the manuscript with a more comprehensive Discussion and Introduction, and revised the text to more clearly state the purpose of the study and its contribution. We are open to the suggestion of changing the manuscript to a full report, as opposed to a Short Report, by bringing in the supplementary results/figures to the main text.

Additional results/analyses:

– Figure 2—figure supplement 1 quantifies behavioral stability

– Figure 3—figure supplement 1 shows that the constrained and concatenated results generalize across all four mice for which there was sufficient data

– Figure 3—figure supplement 2 tests concatenated decoder performance against a null model for drift

– Figure 3—figure supplement 3 Shows that drift is not random, and instead aligns far above change with fast fluctuations in neuronal activity

– Figure 4—figure supplement 1 Shows that the LMS results generalize across animals

– Figure 4—figure supplement 2 Shows that the relative plasticity rates scale with population size, for a fixed error level

– Figure 4—figure supplement 3 Extrapolates the LMS results to a synthetic population of >1000 neurons, showing that very little plasticity would be needed to track the stable subspace as the number of neurons is increased

Reviewer #1:There are a number of theoretical papers (as nicely referenced in this document) about how accurate decoders might be maintained in changing neural populations, but the upside of this work is that:a) The results are taken from experimental data with large enough N's and over enough days that decoding accuracy can be traced, and

We appreciate the reviewer's feedback. We want to point out that we show that *not all drift* sits in a linear subspace. We expand on this in the responses below and in the revised manuscript.

b) This is the simplest of all possible theories of how performance is maintained, and it's reasonably plausible.I have some substantive concerns :1) Given that the consensus decoder had to perform better across days than any single day decoder, it's not clear how surprising these results are.There is nothing to guarantee that a concatenated decoder would perform as well as observed in the data. In fact, taking such concerns onboard, we tested performance against a null model with matched sparsity and within/between day variance. We find that a concatenated decoder performs *substantially better on the data than on this null model*, and extended the analysis to show that this holds across all of the animals that could be analyzedover many days. This is included in a new figure supplement (Figure 3—figure supplement 2).

Secondly, we believe there may be some misunderstanding as to the purpose of constructing these decoders, possibly due to our decision to write a brief manuscript. Our goal is not to predict behavior reliably from data. Our goal is to analyze the dynamics of a drifting representation from the perspective of a system with similar properties and constraints as a downstream neuron/circuit, and then assess, quantitatively, whether these data pose a serious problem for understanding how the brain maintains consistent behavior. The first question we addressed in the paper was indeed a simple, but necessary one: does simple weighted readout work? An affirmative answer suggests a biologically plausible means of reading out the information that is hypothesized to reside in this brain area. The second, immediate, follow on question is: could linear decoding continue to work despite drift? If so, how well, how many units are needed, and does drift induce changes that cannot be confined to a linear subspace? Thirdly, is there a way to quantify the demands placed by drift on connectivity and synaptic plasticity, and do so in a way that is independent of particular models of plasticity? Fourthly, given the actual numbers that emerge from answering the previous questions, is there a specific, parsimonious and biologically plausible model that can find an approximately stable coding subspace and continuously compensate changes that occur outside this subspace? We would argue that none of the follow on questions have obvious answers and all of these questions are important. Reviewer 2 had similar concerns, and we have added Figure 3—figure supplements 1 and 2 address this in more depth. We discuss this in more detail in our response to reviewer 2.

2) It wasn't clear how well this extrapolates across different mice. In some figures, 3 or 4 individuals are compared, others just 2, others yet, just 1 mouse (mouse 4) is mentioned. This is central to the generality of the paper and should be laid out more clearly. Do the concatenated decoder and LMS decoder results hold for more than one individual?

We agree that this was a weakness and we have now addressed it. Overall, we examined five mice, four of which had sufficient neurons recorded for further analyses. We originally focused on two mice (M3, M4) because they had the largest number of tracked days, but the results appear consistent in the other subjects (M1 and M5). We now present supplementary figures for all four mice.

– Figure 3—figure supplement 1 shows that the concatenated decoding results are similar across these four subjects.

– Figure 4—figure supplement 1 shows that the LMS results are general across all four subjects.

3) The arguments about the scaling of the biologically plausible weight adjustments seem a little problematic. It's not clear why the results here form an upper bound on the weight changes needed to maintain accurate decoding.

We agree that this was stated in a glib way and have clarified this point and substantiated it with further analysis. In essence, the argument is that if a biological neuron or circuit had access to even more neurons than we sampled (which we would expect) then the per-synapse rates of change in such a network would certainly be no larger than for a single readout unit with access to a limited population, and would likely be smaller. As the number of useful connections grows, the per-connection contribution shrinks.

We edited the text and added two supplementary figures to better convey how plasticity rate scales with population size.

Figure 4—figure supplement 2 examines scaling with population size in mice 3 and 4. Due to the limited population recorded, this figure does not address scaling to larger populations. Instead, we fix the required error level to match the performance of the full-population LMS (Figure 4—figure supplement 1). We then consider smaller sub-populations, and increase the learning rate to achieve this target error level. Smaller populations require more plasticity to achieve the same decoding performance.

Figure 4—figure supplement 3 extrapolates LMS performance to larger populations (>1000 neurons) by combining neurons from different mice and aligning behavior on each trial. The resulting population exhibits similar scaling relationships as in Figure 4—figure supplement 2. Both weight magnitude and the rate of weight change decrease for larger populations. We also find that the rate of weight change decreases faster than the weight sizes themselves. This confirms that larger (more redundant) populations can be tracked using less per-synapse plasticity.

Secondly, we realised that it might be difficult to directly interpret LMS weight adjustments in the existing model where we impose an upper limit on the change artificially. To simplify things, we removed the limit on the LMS weight change parameter, and control plasticity using only the learning rate parameter *η*. Rather than analyzing the fast fluctuations, we consider only the slow-timescale changes in weights between days, which can be more clearly related (if only qualitatively) to long-term changes in spine sizes or density. These changes are reflected in the revised Figure 4 and associated supplementary figures.

Also, it wasn't clear how the interactions between networks, maintaining congruence, is achieved. That final part of the Discussion was a bit vague.

Thank you, this is useful feedback. We were referring to ideas that are more extensively and clearly articulated in a Current Opinion article that we published last year, which we cite. Nonetheless, our writing in the present manuscript was vague and we have rewritten this paragraph in the Discussion.

We believe that the revised Discussion better emphasizes the insights into drift in PPC population codes provided by our analysis, and more clearly states the questions we addressed. We have also re-written the Discussion to more clearly convey the limitations of our study, and to highlight new experimental and theoretical directions suggested by our results.

Reviewer #2:Loback et al. re-analyze data from Driscoll et al., 2017, which had previously shown that PPC representations are unstable over days during a delayed VR T-maze task. Here, using linear decoders, they find that a static decoder can do a reasonable job if trained on data from all days, and that an old model of synaptic weight updates can be applied to maintain decoder performance. The analyses seem to have been done reasonably, but the results strike me as rather shallow and are based on limited data.

We appreciate the constructive comments, other reviewers noted similar concerns. We have substantially revised the text and extended the manuscript with deeper analyses, extended across animals. We believe that this revised manuscript addresses these concerns.

We've added Figure 3—figure supplement 1 and Figure 4—figure supplement 1) to show that the results are generalize over all four mice considered. Please see our response to reviewer 1, which goes into greater depth regarding results from additional subjects.

We also emphasize that we chose an older and simple model of plasticity (LMS) after considerable deliberation and exploration. The choice was not ad-hoc because our goal was not to invent yet another model of plasticity, but instead to evaluate how difficult the problem of drift would be for a simple and biologically plausible learning rule. We evaluated several decoders and learning rules, including nonlinear methods, Gaussian process methods, etc. In all cases, more sophisticated methods required additional assumptions about the mechanism of plasticity and obscured any biological interpretation.

Although it was sometimes possible to get better decoding performance with more sophisticated approaches, this was not our goal. We felt that LMS was more appropriate for lower-bounding the required plasticity to achieve a target decoding performance. The simplicity (and limitations) of LMS made it a useful assay for determining how disruptive drift would be in a biological system. Our reasoning was that if a relatively under-powered local learning rule could track drift, then it would also be very likely that neurons in the brain could do the same (or better), especially with access to a larger PPC population.

We therefore do not believe that the choice of a simple, widely known and parsimonious model is a weakness, but rather a strength. There was no guarantee that such a simple model would work and the fact that is does is important given the fundamental questions raised by the experimental observations.

The first main result is simply that unstable representations cause single-day (linear) decoders to generalize poorly, but a multi-day decoder to perform somewhat better. I have two issues with this result:1) Given that activity is sparse and does not have systematic shifts in tuning, this decoding result is very nearly a mathematical necessity. Because of sparsity, the decoder likely ends up built so that different units drive the decoder performance on different sessions. This would not be news. Further, there are no null models analyzed for what would happen under different patterns of tuning shifts, which would be a helpful comparison. It is therefore not clear whether there is anything to be surprised at here.

We have taken on board this concern, especially the issue of sparse activity. We constructed a null model which we now present in Figure 3—figure supplement 2 and associated text in Results. In fact, the performance of a concatenated decoder is far above chance compared to a null model with matched variance, sparsity and random drift. Please also refer to our response to the similar issue raised in reviewer 1's first comment, especially our clarification on the purpose of this study and the non-obvious questions it addresses.

We also now present further evidence that drift in the data is not random (Figure 3—figure supplement 3) and associated text in Results. The overlap of drift with behavior coding directions is significantly above chance, but a significant proportion of drift still lies in the null space for location coding.

Overall, our results now provide deeper insight and show that drift (partially) preserves important features of population tuning curve statistics. In light of this, we now feel that the original result is stronger: drift is constrained in a way that could make it more disruptive than chance, but we find that a stable subspace exists nonetheless.

After addressing these issues we feel even more strongly that these results are important for the community, especially since several other groups are now examining drift and stability in other brain areas and other model organisms.

2) How stable is the behavior within and across sessions? Details of behavior matter all over the brain (Stringer, Pachitariu et al., 2019, Musall, Kaufman et al., 2019), so it is possible that drift in behavioral details could lead to these shifts as well. At least, it should really be shown whether the parameters the authors track are stable over time.

Yes, we agree. Driscoll et al., 2017 verified that the overall task performance was stable, but behavioral details are also important.

To address this, we have added Figure 2—figure supplement 1. We assessed behavior changes over time, and found systematic changes only in the forward movement of mouse 4. In all other instances we found no systematic changes. While statistically significant, daily fluctuations in behavior were small. Importantly, all behavioral statistics recorded were stable for three of the four mice studied, suggesting that our results are general and do not arise from systematic changes in behavior.

We now refer to this figure in the main text:

"Behavioral variables were stable over time with some per-session variability (mouse 4 exhibited a slight decrease in forward speed over two weeks; Figure 2—figure supplement 1)."

These points said, there is value in this section. The quantifications of instability and of how many neurons are required for good decoder quality are helpful, and the point that only 6% of neurons are even in the top 50% of informativeness is surprising and interesting.The second major result is the application of the Widrow and Hoff, 1962 model. If I understand correctly, this is primarily a different way of quantifying how fast the tuning changes occur (requiring ~2%/minute weight changes), and secondarily a proof of principle for that model. However, unless I'm missing something, this is a one-mouse result. That would not meet the standard for the field. In addition, comment #2 above applies to this result as well, making it harder to interpret.Before addressing the LMS issue, we want to point out that *the second major result is a quantification of how much drift occurs outside a linear subspace. We find that a non-negligible component does indeed lie outside a linear subspace, thus preventing long term, reliable decoding by a fixed decoder*. Before attempting to find an example of a biologically plausible model that could compensate for this, we quantified the expected per-synapse adjustment that would be required to compensate for this component of drift independently of a specific learning rule. This result and analysis is in Figure 3D-E.

Turning to the issue of the LMS results, we agree that showing only one example was a weakness. We now provide results from LMS from all mice (Figure 4—figure supplement 1); the results generalise.

Reviewer #3:Overall, I find that the paper is clearly written and everything seems technically correct. However, I also find that it lacks scientific novelty. While I find the idea of linking the observed reorganization of neural activity with synaptic plasticity exciting, I find that the paper does not quite achieve that. I think the authors would need to work out some concrete consequences/constraints on plasticity forthis paper to become viable.We appreciate this assessment, and share the reviewer's excitement regarding linking neural activity with synaptic plasticity. We need to immediately point out that this was not the sole aim of the paper. *The other important aim was to characterise drift as it occurs experimentally and ask if drift is any way structured or minimally disruptive with respect to a plausible readout mechanism.* In doing so we are directly testing a well-known theoretical proposal that 'irrelevant' changes in a neural code can be confined to a null space. Our conclusions to this crucial question are more fully discussed elsewhere in this response and we have revised the manuscript substantially to further articulate them.

Turning back to the problem of relating drift to plasticity, the reviewer will appreciate that it is very difficult to directly connect population activity to synaptic plasticity. Where do we start? How do we avoid making too many assumptions and at the same time provide concrete, interpretable models and numbers that can be directly related to the biological system?

We feel that our revised analysis goes a long way to achieving this by considering several variations of a "plausible worst case" scenario that addresses the most pressing question raised by the data, namely, does activity drift pose an immediate problem for understanding the function of PPC and other cortical circuits? We would argue that our analysis does provide concrete constraints and consequences for plasticity, not just qualitatively, but down to actual numbers that are meaningful given known physiology and connectivity.

Specifically:

– We now quantify the effect of population size on long term decoder performance

– We now quantify how much drift occurs in a linear subspace, finding that a non-negligible component does not reside in a linear subspace, and will eventually degrade a fixed readout to chance levels

– We quantify the extent to which a 'best subset' of neurons exists in the population, finding that this subset turns over completely and surprisingly rapidly

– We find a way to estimate how much plasticity would be required, under reasonable and clear assumptions, to compensate for drift *independently of a plasticity mechanism* – We provide a parsimonious, biologically plausible example of a specific learning rule that can, indeed, achieve this compensation

We would urge the reviewer to contemplate what an alternative approach would consist of that could better address these issues. This is not to say that our original manuscript did not have weaknesses. We failed to include analyses across animals and didn't go as deep as we could have in the analyses that we performed. We also didn't fully articulate the main questions and goals of the study in the very terse manuscript we originally submitted, so some of the above contributions were easy to overlook. We have addressed this as well as adding new results, as detailed below in this response. We believe the manuscript is now clearer and stronger.

Broadly speaking, the current study is divided into two parts. The first part is a re-analysis of the data of Driscoll et al., 2017, which is performed in Figures 1-3. The authors use decoding methods to retrieve task information from the population activity. While some of the details of the population decoding methods are different to those used by the Driscoll et al., the overall conclusions are the same. The strongest point of the re-analysis is that the authors more clearly quantify the strength of the day-to-day changes using decoders that are constrained to change only little over days. That is a nice twist that was not performed in the Driscoll paper.The second part of the paper is an attempt to relate these day-to-day changes to synaptic plasticity (Figures 3 and 4). This part is rather brief and quite sketchy. Roughly, the authors simply reformulate the constrained decoder as an adaptive decoder. Conceptually, that is similar to the ideas brought forward by Rokni et al., Ajemian et al., and others. What could make this part interesting, is if this link could be made stronger, i.e., if it could really be a link to synaptic plasticity, rather than a link to a hypothetical readout. But even if the authors limit themselves to a single readout neuron, many questions are left unaddressed, e.g. how to extrapolate the adaptation rules for the decoder to realistic network sizes.

Although we cannot access synaptic plasticity directly in these data, we feel that our decoding-based analysis can provide a useful approach for studying constraints on plasticity from recordings of population activity alone.

As outlined in more detail in our responses to reviewers 1 and 2, we have added several new supplementary analyses. These analyses show that the results are general across subjects (Figure 3—figure supplement 1, Figure 4—figure supplement 1), and that observed drift is structured (Figure 3—figure supplements 2, 3). Drift aligns with neural activity, especially noise correlations (Figure 3—figure supplement 3).

We also now address scaling of these results with network size. In Figure 4—figure supplement 2 we show that the required rate of plasticity increases for smaller network sizes, for two mice (M3, M4).

Extrapolating the results to large networks was more challenging, but we were able to construct a synthetic population by aligning trials from different mice in pseudo-time (Figure 4—figure supplement 3). Although this analysis extends over only 6 days, it scales to >1000 neurons and shows that the required plasticity continues to decrease as more neurons are added.

We feel that this report is useful for the community, as many groups are beginning to study drift and plasticity in other brain areas and in other model organisms. We feel that the decoding-based approach to drift is a useful foundation, and that our results will contribute to further experimental and theoretical work on this topic.

Overall, we would summarize the contributions of our revised work as follows:

– While there has been speculation on how to reconcile stable representations with drift in neuronal tuning, this study tests these ideas against experimental data.

– Our work highlights that drift must be structured if it is to preserve population-coding statistics, and our analysis shows that drift dynamics are indeed structured far above chance.

– We find that drift consists of daily fluctuations around a more stable substructure, which nevertheless changes over weeks to months.

– We find that some, but not all, drift occurs in a linear coding subspace. This has immediate implications for existing theories of circuit function.

– Our modelling demonstrates that this structured drift could allow a readout neuron to readily compensate for changes in the neural code, and quantifies the constraints on plasticity and connectivity independently of specific learning rules while also providing a specific example of a plausible model that can operate within these constraints.

– Our results motivate further experiments to search for neural correlates of error signals between brain areas, which we believe would be required to maintain consistency between drifting representations.

– Our results also motivate future theoretical treatment of the underlying cause of drift, how it is related to plasticity, learning and biological noise and whether it is expected to be a universal feature of large, adaptive neural circuits.

Other comments:1) It was not clear to me what happens with the decoders within a session and between days. Do decoders 'jump' between days or stay roughly the same? How does that influence the adaptation rules?

Agreed. We changed the plotting code so that Figure 4B, Figure 4—figure supplement 1, and figure 4—figure supplement 3 to show discontinuities between days. Per-day fluctuations are present, and can sometimes even lead to improvements across days.

Overall, sharp "jumps" in the LMS error are rare, since LMS tracks close to optimal performance.

2) Legend of Figure 4 and subsection “Biologically plausible weight adjustment can compensate for ongoing reconfiguration of PPC activity”. You repeatedly state that you approach the 'concatenated decoder.' I guess that should be the 'constrained decoder', otherwise it makes no sense to me.

Thanks; we have changed the caption in Figure 4 to read:

"a decoder trained over all testing days".

[Editors’ note: what follows is the authors’ response to the second round of review.]

[…] Points left to address in full:1) Please go back and consider the more interesting null model in the other analyses and quantifications in this manuscript. This will improve many other parts of the paper. Please also place this new null model result in the main text of the paper.

1) We have included a new null model in the main figures with sparsity matched to the data (Figure 3D); this supersedes the original null model and it is discussed in further detail in response to Points 2 and 3 below.

2) We have used the null model to evaluate levels of plasticity and performance of local ongoing drift compensation with the LMS algorithm.

3) We have constructed a new rank-constrained model of drift in Figure 4—figure supplement 1, which quantifies the level of constraint needed to best match drift to data.

4) We have evaluated null models for head direction and velocity; results were similar but we have omitted the quantification because would add numerous figure panels without adding any insight.

5) We have added text interpreting and discussing the new null models in the main text.

2) Regarding the new null model:Past evidence has clearly shown that neural tuning (or population activity) changes both randomly (assumed to be due to plasticity noise) and directionally (assumed to be due to feedback and learning). With the new null model, the analyses attempts to rule out a random walk. This is valuable effort. However, please add commentary on how this null model is useful despite ignoring the influence of the systematic, directional changes which were already demonstrated in the past, including the authors' own data, and which have usually been related to ongoing learning.

We have designed and analyzed a new variation of the (sparsity matched) null model that constrains drift to low rank subspaces and quantifies how rank affects the degradation of the code with respect to static readout weights. This is presented in a new Figure 4—figure supplement 1. As discussed in the revised manuscript, this new analysis shows that drift in the data can be quantified in terms of both a random and a systematic component and that drift is far more systematic than would be expected by chance. By modelling drift as confined to a subspace we are now able to provide and interpret a measure of how systematic the drift is in terms of the subspace rank that best matches the data.

We note that the review comments here neglect the alignment analysis in the previous version of the manuscript (now in main Figure 4C, D), which again shows evidence of (and quantifies) the systematic and random components of drift.

We have now extensively modelled, analyzed, interpreted and discussed systematic vs. random drift. Nonetheless, the null model in Figure 3D is useful precisely because it omits systematic changes in the population code. As we outline in the manuscript, the purpose is to illustrate that random, diffusive drift would rapidly degrade a downstream readout. The fact that the null model performs worse than the data confirms that the systematic structure present in the drift makes it far less destructive to a linear readout than expected by chance. The modifications we have made in the revision also now shows that sparsity doesn’t make this finding trivial. Moreover, we still see a slow degradation of decoding within the data, which motivates the later analyses that quantify how much additional plasticity would be required of a downstream area. For the data we have, these analyses together show that in the long run, regardless of a systematic component, drift degrades an optimised static linear readout, indicating a need for ongoing plasticity.

Finally, to clarify once more: this data was explicitly gathered not during ongoing learning. Behavioral performance had plateaued before imaging began. Any additional change in neural activity is not a feature of measurable behavioral improvement. We posit that systematic changes in activity are a feature of the maintenance of learned behaviors. The original 2017 paper alluded to this ideas but came short of demonstrating them in the analysis. The decoders used in the original paper simply demonstrated the utility of using a large number of cells for decoding a single binary variable: trial type (i.e. decoding a single bit of information). It is highly unlikely that the activity in PPC amounts to only 1 bit. The reviewers will therefore recognise that the success of decoding a single bit doesn’t say much, if anything, about the **extent** to which drift damages or preserves information in a static readout, it simply says that drift doesn’t completely destroy a small amount of information within a limited timescale. It also doesn’t say anything about how readout weights might be learned/maintained.

3) Please address in full the expanded review comments sent during the initial appeal. That text is reproduced here:Thank you for sending us your thoughts and questions about the reviewer comments. […] Without that, we would have expected that the concatenated decoder would work well.

It turns out that the reviewer’s assertion about the concatenated decoder is incorrect. In the new null model we now match the sparsity of the activity in the data using the activation patterns themselves. A comparison with the previous (non-sparse) null model is shown in Author response image 1.

We see that the degradation of a concatenated decoder on the sparsity-matched null model is in fact more severe than the original null model. This shows, contrary to the reviewers' intuition, that ‘sparse’ representation of the task variables does not make existence of a multi-day decoder trivial. In fact, it makes its existence statistically less likely.Intuitively, this is because the type of sparseness in the data corresponds to only a small number of cells representing a given range of a task variable (e.g. a handful of cells active in a particular velocity range). Any drift that affects a significant proportion of these cells cannot be compensated by other cells in the population, unlike in a non-sparse case where a given cell may have activity spread over a large range of task space.

Nonetheless, the reviewer’s challenge was useful because it prompted us to construct a more relevant null model which shows that the structure of drift in the data is even less likely to occur by chance than one might suppose.

Second, as it was understood by the reviewers, the manuscript argues that we should be surprised that updating decoder weights with the Widrow and Hoff model works here. From Driscoll, 2017, we have an idea of how rapidly location selectivity changes, and how rapidly a static decoder decays. Given that we know this, how rapidly would you expect to have to change the decoder? We didn't see much in the paper that wasn't just a different way of quantifying the same tuning changes. One reviewer suggested adding more specific null models because they think this would let the authors answer these deeper questions. For example, are all of the neurons smoothly changing their tuning? Do some change fast and others slow, and is this a continuous distribution? Is there coordination between neurons' tuning changes or are neurons changing independently? The current null models are extremes: the shuffle is related to a model where everything changes instantly (obviously wrong), and the same-day decoder is equivalent to there being no changes ever (which we know is wrong from Driscoll, 2017). So, what new have we learned?

First we stress that these results aren’t the only two contributions of this study; we have enumerated the key contributions below.

Addressing this point, we have now used the (new) null model of drift to assess how well online compensation in the Widrow-Hoff LMS algorithm might be expected to perform. We find that performance of LMS on a sparsity-matched null model of drift is substantially worse than the data. Thus, the results in this section are far from trivial and cannot be taken for granted. We have quantified these results in the text that accompanied Figure 5:

"These results suggest that small weight changes could track representational drift in practice. […] This further indicates that drift is highly structured, facilitating online compensation with a local learning rule."

Finally, we would remind the reviewers that it is one thing to have a hunch or suspect that something may be possible. It is quite another to explicitly demonstrate it and to find a means for doing so. We thus believe the main value of this specific result is not that it has some kind of ‘shock’ value, but that it is a principled and informative scientific analysis: we established a way to place theoretical bounds on levels of plasticity required to compensate drift, independently of any learning rule (Figure 4); we then showed they could be achieved using a biologically plausible learning rule (Figure 5). Neither of these steps is obvious or trivial. Both are meaningful.

Finally, regarding the result that only 6% of neurons are in the top 50% all 10 days: again, we lack the context to know how surprised we should be. If we suppose that the informative neurons are chosen randomly each day, then we'd expect the number of neurons that are in the top 50% for 10 days to be 0.5 ^ 9 = ~0.2%. In that case, 6% is surprisingly high. Looking at Driscoll's Figure 2B, ~40% of neurons keep their place preference for 10 days. In that case, 6% is surprisingly low. In fact, why is it so low? Could this just mean the decoder is under-regularized?

We used regularization in the linear models. This is now documented fully in the Materials and methods and does not affect the ranking of the ‘best subset’ of cells.

What’s happening here is the following: neurons with stable tuning peaks can exhibit unstable signal-to-noise ratios. In other words, the location of the maximum firing may change little, but the profile of firing away from the peak can change a lot. As a result, decoders that try to rely on previously good or stable cells eventually suffer when these cells become less reliable.

Therefore means of assessing tuning curve stability used in Driscoll et al. is not the correct measure for assessing stability with respect to a downstream neuron with fixed synaptic weights. This highlights the importance of the decoding perspective in Loback et al. We now clarify this in the text:

"For all subjects, no more than 1% of cells were consistently ranked in the top 10%, an no more than 13% in the top 50%. We confirmed that this instability was not due to under-regularization in training (Materials and methods: Best K-Subset Ranking)."

This instability might seem surprising, since Driscoll et al., 2017, found that ∼40% of cells were tuned to similar preferred locations over time. We find that even this ‘stable’ subset exhibited daily variations in their Signal-to-Noise Ratio (SNR) with respect to task decoding. For example, no more than 8% of neurons that were in the top 25% in terms of tuning-peak stability were also consistently in the top 25% in terms of SNR for all days. If a neuron becomes relatively less reliable, then the weight assigned may become inappropriate for decoding.